# FAST COMPUTATION OF GAUSSIAN PROCESSES AUGMENTED BY SYNTHETIC SIMULATOR DATA

## ABSTRACT

When the amount of training data is limited, augmenting it with generated data from a simulator can be a beneficial approach to improving prediction accuracy. However, there are no clear metrics on which generated data should be added to the training set and in what proportion, especially when the predictive model is a Gaussian Processes (GPs) model. To address this, we propose using the log marginal likelihood as a guiding metric. The log marginal likelihood is a theoretically grounded criterion for model selection when incorporating simulator-generated data into the training set. Nevertheless, computing this metric for GPs is computationally expensive. To overcome this challenge, we introduce a faster method for calculating the log marginal likelihood by considering the Cholesky factor and matrix element dependencies. Experimental results demonstrate that metrics utilizing the log likelihood outperform basic methods in mean squared error on test set.

## 1 INTRODUCTION

Gaussian Processes (GPs) are widely used as predictive models due to their flexibility as non-parametric models and their ability to output confidence levels for predictions. Like other machine learning models, GPs require training data, but in real-world applications, sufficient training data is often lacking.

When training data is limited, if access to a simulator capable of generating training data is available, incorporating the generated data can be a beneficial approach to improving prediction accuracy. Typical data augmentation techniques, such as image flipping and trimming, are a part of this approach, but here we consider more complex simulators. For example, when predicting solar power generation from weather conditions, a power generation simulator can be used in addition to the previous year's data.

However, the data generated by simulators often deviates from the true distribution. In the solar power generation example, the inverters and other equipment used in the simulator tool may differ from those actually used by the company that wants to predict power generation.

Several studies have investigated the utilization of simulators that partially deviate from the true distribution for training predictive models. These approaches can be broadly classified into two categories: modifying the objective function of the predictive model or modifying the generated data added to the predictive model. The former incorporates a penalty term for the degree of deviation from the simulator into the loss function, with a representative example being physics-informed neural networks Raissi et al. (2019). However, such objective function modification methods are difficult to apply to non-parametric models, including GPs, because non-parametric models do not have parameters to optimize with respect to the objective function (except for kernel hyperparameters). The latter approach of modifying the generated data added to the predictive model is applicable to non-parametric models such as GPs, and our method belongs to this category. Within this approach, existing research focuses on Auto Data Augmentation Cubuk et al. (2019; 2020); Ho et al. (2019); Lim et al. (2019), which parameterizes data augmentation and optimizes those parameters (for more details, see Appendix A.4). However, while Auto Data Augmentation can be applied to simulators where individual operations can be parameterized, such as image flipping and trimming, it is difficult to apply to off-the-shelf simulation tools where machine learning engineers cannot access the internal workings and thus cannot parameterize individual operations.

Therefore, we adopt a framework that evaluates each generated data point and decides whether to include it in the training data. This framework is applicable to non-parametric models since it does not involve updating model parameters, and it is also applicable to off-the-shelf simulation tools that cannot be internally modified, as it does not require parameterizing the simulator.

We propose using the log marginal likelihood on training data for evaluating generated data. The negative log marginal likelihood is a metric that measures the model's fit to the training data and has a theoretical foundation that it matches, on average, the Kullback–Leibler (KL) divergence between the true distribution and the model's distribution. For more details, see Appendix F, G and Shlens (2007).

Consequently, the proposed method of evaluating generated data using the log marginal likelihood effectively adds the generated data to the training data if the inclusion of the generated data brings the GPs' predictive distribution closer to the true distribution on average, and does not add it otherwise.

Furthermore, evaluating each candidate training data point using the negative log marginal likelihood can be time-consuming, so we propose a method for fast computation by considering the Cholesky update and the dependencies between matrix elements. Specifically, using our method, the computational cost to select generated data to add to the training set can be reduced from $O(M^3N + M^2N^2 + MN^3)$ to $O(M^2N + MN^2)$ where $N$ is the number of true training data and $M$ is the number of data generated by the simulator.

The contributions of this research are 2 folds. First, we proposed using the negative log marginal likelihood of the GPs[1] as a criterion when selectively adding simulator-generated data to the training data. Second, we proposed an algorithm to efficiently compute the negative log marginal likelihood when selecting training data (Section 4.2).

## 2 RELATED WORKS

### 2.1 EFFICIENT TRAINING DATA SELECTION METHODS FOR GPS

GPs have a computational complexity of $O(N^3)$ during prediction, where $N$ is the number of training data points. Therefore, two approaches have been developed to reduce the amount of training data as much as possible. One approach is called Subset of Data (SoD), which involves retaining only the good data points from the training data and discarding the rest. The criterion for goodness is diversity, and various ways of measuring diversity have been proposed (see Appendix A.1 for more details). However, selecting generated data from the simulator to increase diversity may also include data points that deviate from the true distribution. We verify this experimentally. The second approach is sparse Gaussian processes, which generate a small number of pseudo-training data points (see Appendix A.2 for more details). However, this approach is not closely related to the objective of selecting generated data from the simulator.

### 2.2 COMPUTATIONAL TECHNIQUES FOR GPS

Due to the high computational complexity of GPs, various computational techniques have been developed. After training with $N$ data points (after computing the inverse matrix with $N$ data), an additional data point can be learned with an added computational effort of $O(N^2)$ using a technique called rank-one update Nguyen-Tuong et al. (2008); Seeger (2004). This technique is used in part in our method (see Appendix A.3 for more details). Additionally, methods for efficiently computing the Cholesky factor of the kernel matrix have been proposed Osborne (2010). However, to efficiently compute the log marginal likelihood, it is necessary to compute the Cholesky factor of the covariance matrix (see footnote 5 for mathematical explanations). Section 4.2 presents our proposed method to address this issue.

## 3 METRICS

The metric for deciding whether to add generated data from the simulator to the training data should be based on whether the addition improves the prediction accuracy of the GPs. We use the log

---

[1]The algorithm we proposed is specialized for regression models.

marginal likelihood of the GPs as a metric, which is the average KL divergence between the predictive distribution and the true distribution. (For an explanation of the approximation, refer to Appendix F) Therefore, if the log marginal likelihood improves after adding the generated data, it means that the predictive distribution of the GPs has moved closer to the true distribution on average, and the prediction accuracy is expected to increase.

The process of the proposed method is straightforward. First, the hyperparameters of the GPs are learned using training data. Next, data $(\mathbf{x}, y)^2$ is sampled from the simulator and added to the training data of the GPs. We measure the log marginal likelihood of the GPs, and if it improves, we accept the sampled data as valid training data, otherwise, we discard it. This process is repeated until no continuous improvement is observed or until all generated data has been examined. The GPs that possess both the original training data and the accepted generated data serve as our final prediction model.

In Section 3.1, we derive the marginal likelihood when adding the training data candidates generated from the simulator. In Section 4, we propose a method to quickly compute that marginal likelihood.

## 3.1 Marginal Likelihood of GPs when Generated Data is Added

We define the symbols as follows: $\mathbf{x} \in \mathbb{R}^d$ is a random variable, $\mathbf{X}^N = (\mathbf{x}_1, \mathbf{x}_2, \ldots, \mathbf{x}_N)$ are $N$ independent random variables following the same distribution, $y \in \mathbb{R}^1$ is a random variable, and $\mathbf{y}^N = (y_1, y_2, \ldots, y_N)$ are $N$ independent random variables following the same distribution. Let $(\mathbf{X}^N, \mathbf{y}^N)$ be the training data and $(\mathbf{X}^{m^*}, \mathbf{y}^{m^*})$ be the $m$ training data candidates generated from the simulator. Note that $M$ is the total number of data generated from the simulator, and $m$ is the number of training data candidates generated by the simulator up to the current step, as data is greedily added to the training set. We also denoted the star in $(\mathbf{X}^{m^*}, \mathbf{y}^{m^*})$ to explicitly indicate that it is not a sample from the true distribution.

As we mentioned at the beginning of Section 3, in order to evaluate samples from the simulator, we determine the marginal likelihood of the GP when such a sample is added. The negative log marginal likelihood (the free energy) of the discriminative model when generated data is added was $F_{m^*} = -\log p(\mathbf{y}^N | \mathbf{X}^N, \mathbf{y}^{m^*}, \mathbf{X}^{m^*})$. (Refer to Appendix G for the derivation.) In the case of GPs, since the predictive distribution can be analytically determined (see Appendix H), the marginal likelihood $p(\mathbf{y}^N | \mathbf{X}^N, \mathbf{y}^{m^*}, \mathbf{X}^{m^*})$ is also trivially obtained. To define the notation, let's express the joint distribution of generated data and training data as follows:

$$
\begin{pmatrix} y_{1^*} \\ \vdots \\ \vdots \\ y_{m^*} \\ y_1 \\ \vdots \\ \vdots \\ y_N \end{pmatrix} \sim \mathcal{N} \left( \begin{pmatrix} 0 \\ \vdots \\ 0 \\ 0 \\ \vdots \\ 0 \end{pmatrix}, \begin{matrix} \mathbf{x}_{1^*} \\ \vdots \\ \mathbf{x}_{m^*} \\ \mathbf{x}_1 \\ \vdots \\ \mathbf{x}_N \end{matrix} \begin{pmatrix} \overset{\mathbf{x}_{1^*}{}^3 \cdots \mathbf{x}_{m^*}}{\boxed{\begin{matrix} \mathbf{K}_{m^*} + \sigma^2 \mathbf{I}^4 & \mathbf{K}_{N,m^*} \\ \hline \mathbf{K}_{N,m^*}^{\mathsf{T}} & \mathbf{K}_N + \sigma^2 \mathbf{I} \end{matrix}}} \end{pmatrix} \right). \tag{1}
$$

Here, each $\mathbf{K}$ is the kernel of the corresponding rows and columns of $\mathbf{x}$. Using this notation, the free energy when generated data is added as training data is expressed as $p(\mathbf{y}^N | \mathbf{X}^N, \mathbf{y}^{m^*}, \mathbf{X}^{m^*}) = \mathcal{N}(\mathbf{K}_{N,m^*}^{\mathsf{T}} [\mathbf{K}_m + \sigma^2 \mathbf{I}]^{-1} \mathbf{y}^{m^*}, [\mathbf{K}_N + \sigma^2 \mathbf{I}] - \mathbf{K}_{N,m^*}^{\mathsf{T}} [\mathbf{K}_m + \sigma^2 \mathbf{I}]^{-1} \mathbf{K}_{N,m^*})$.

Next, as a metric to determine whether to add the $m+1$-th data to where $m$ pieces of generated data have been adopted, we will explain the free energy when the $m+1$-th generated data is added. Let's

---

[2]The definition of $(\mathbf{x}, y)$ is provided in Section 3.1.

[3]$\mathbf{x}_{1^*} \ldots \mathbf{x}_{m^*}, \mathbf{x}_1 \ldots \mathbf{x}_N$ represent the input variables for the kernel function that constructs the covariance matrix.

[4]Different-sized identity matrices appear in the paper, but the size is easily inferred from the context, so they are all uniformly denoted as $\mathbf{I}$.

define the notation of the covariance matrix by adding $\mathbf{x}_{m+1*}$ to Equation 1 as follows:

$$
\begin{pmatrix} y_{1*} \\ \vdots \\ y_{m*} \\ y_{m+1*} \\ y_1 \\ \vdots \\ y_N \end{pmatrix} \sim \mathcal{N} \left( \begin{pmatrix} 0 \\ \vdots \\ 0 \\ 0 \\ 0 \\ \vdots \\ 0 \end{pmatrix}, \begin{array}{c} \begin{matrix} \mathbf{x}_{1*} \\ \vdots \\ \mathbf{x}_{m*} \\ \mathbf{x}_{m+1*} \\ \mathbf{x}_1 \\ \vdots \\ \mathbf{x}_N \end{matrix} \end{array} \begin{pmatrix} \overset{\mathbf{x}_{1*}\cdots\mathbf{x}_{m*}\,\mathbf{x}_{m+1*}}{\mathbf{K}_{m+1*} + \sigma^2\mathbf{I}} & \overset{\mathbf{x}_1\cdots\mathbf{x}_N}{\mathbf{K}_{N,m+1*}} \\ \hline \mathbf{K}^{\mathsf{T}}_{N,m+1*} & \mathbf{K}_N + \sigma^2\mathbf{I} \end{pmatrix} \right). \quad (2)
$$

At this time, the free energy can be simply extended as: $F_{m+1*} = -\log p(\mathbf{y}^N|\mathbf{X}^N, \mathbf{y}^{m^*}, \mathbf{X}^{m^*}) = -\log\mathcal{N}(\mathbf{K}^{\mathsf{T}}_{N,m+1*}[\mathbf{K}_{m+1*} + \sigma^2\mathbf{I}]^{-1}\mathbf{y}^{m+1}, [\mathbf{K}_N + \sigma^2\mathbf{I}] - \mathbf{K}^{\mathsf{T}}_{N,m+1*}[\mathbf{K}_{m+1*} + \sigma^2\mathbf{I}]^{-1}\mathbf{K}_{N,m+1*})$.

For subsequent sections, let's expand the content of the free energy as follows:

$$
F_{m+1*} = \frac{1}{2}\left(\mathbf{y}^N - \boldsymbol{\mu}_{m+1}\right)^{\mathsf{T}}\boldsymbol{\Sigma}^{-1}_{m+1}\left(\mathbf{y}^N - \boldsymbol{\mu}_{m+1}\right)\frac{1}{2}\log|\boldsymbol{\Sigma}_{m+1}| - \frac{N}{2}\log 2\pi, \quad (3)
$$

$$
\boldsymbol{\mu}_{m+1} = \mathbf{K}^{\mathsf{T}}_{N,m+1*}[\mathbf{K}_{m+1*} + \sigma^2\mathbf{I}]^{-1}\mathbf{y}^{m+1}, \quad (4)
$$

$$
\boldsymbol{\Sigma}_{m+1} = [\mathbf{K}_N + \sigma^2\mathbf{I}] - \mathbf{K}^{\mathsf{T}}_{N,m+1*}[\mathbf{K}_{m+1*} + \sigma^2\mathbf{I}]^{-1}\mathbf{K}_{N,m+1*}. \quad (5)
$$

The overall procedure of the algorithm is to randomly draw a generated data in sequence and formally add the $m+1$-th generated data to the training data if $F_{m+1*} < F_{m^*}$, and discard it otherwise. Section 4 will explain an algorithm to perform this evaluation quickly.

## 4 REDUCING THE COMPUTATIONAL COST OF FREE ENERGY UPDATE

In order to rapidly compute equation 3, there are two challenges. The first one is to compute $\mathbf{K}^{-1}_{m+1*} + \sigma^2\mathbf{I}$ in the mean (Equation 4). Though the computation of $\mathbf{K}^{-1}_{m+1*} + \sigma^2\mathbf{I}$ can be efficiently determined through the Cholesky decomposition $\mathbf{K}_{m+1*} + \sigma^2\mathbf{I} = \mathbf{L}_{m+1}\mathbf{L}^{\mathsf{T}}_{m+1}$ (where $\mathbf{L}_{m+1}$ is an $(m+1) \times (m+1)$ lower triangular matrix), the computational cost of obtaining $\mathbf{L}_{m+1*}$ still remains $O(m^3)$. Hence, the total computational cost, even if generated samples are adopted every time, amounts to $O(M^4)$ for incorporating $M$ data points. This makes the computation challenging. However, by utilizing $\mathbf{L}_m$ from the previous step, the computation of $\mathbf{L}_{m+1}$ can be achieved in $O(m^2)$, and the total computational cost can be kept within $O(M^3)$. This technique is called Cholesky Update Osborne (2010). For other existing acceleration techniques, refer to Appendix A.3. The second one is that the inverse matrix of the free energy variance-covariance matrix, $\boldsymbol{\Sigma}^{-1}_{m+1}$, appearing in the first term on the right side of Equation 3, requires a matrix multiplication cost of $O(m^2N + mN^2)$ and $O(N^3)$ for the inversion even when the efficiently updated $\mathbf{L}_{m+1}$ is used Osborne (2010). Thus, the total amounts to $O(M^3N + M^2N^2 + MN^3)$, exceeding the allowable range. However, we propose a new algorithm which passes through the Cholesky decomposition of the variance-covariance matrix $\boldsymbol{\Sigma}_{m+1} = \mathbf{V}_{m+1}\mathbf{V}^{\mathsf{T}}_{m+1}$, and reuses $\mathbf{V}_m$ from the previous step for calculating $\mathbf{V}_{m+1}$[5]. The algorithm achieves a computational complexity of $O(mN + N^2)$ per step, with a total cost of $O(M^2N + MN^2)$, keeping it within the quadratic order for N. Once the Cholesky factor $\mathbf{V}_{m+1}$ is determined, the second term $|\boldsymbol{\Sigma}_{m+1}|$ in Equation 3 can also be immediately computed.

### 4.1 COMPUTING $\mathbf{K}^{-1}_{m+1*} + \sigma^2\mathbf{I}$

First, we explain how to efficiently compute the Cholesky factor $\mathbf{L}_{m+1}$ of $\mathbf{K}^{-1}_{m+1*} + \sigma^2\mathbf{I}$ from $\mathbf{L}_m$. $\mathbf{K}_{m+1*}$, as shown in Equation 6, possesses a structure extended by the kernel vector $\mathbf{k}_{m+1*}$ and

---

[5]Osborne (2010) proposed an efficient method to compute $\mathbf{L}_{m+1}$ from $\mathbf{L}_m$ in $O(m^2)$ for the Cholesky factor $\mathbf{K}_{m+1*} + \sigma^2\mathbf{I} = \mathbf{L}_{m+1}\mathbf{L}^{\mathsf{T}}_{m+1}$. In contrast, we propose an efficient method to derive $\mathbf{V}_{m+1}$ from $\mathbf{V}_m$ in $O(mN + N^2)$ for $\boldsymbol{\Sigma}_{m+1} = \mathbf{V}_{m+1}\mathbf{V}^{\mathsf{T}}_{m+1}$.

scalar $k_{m+1*}$ due to the added generated data $\mathbf{x}_{m+1*}$ to the covariance matrix $\mathbf{K}_{m*}$ prior to the addition:

$$
\mathbf{K}_{m+1*} + \sigma^2 \mathbf{I} = 
\begin{array}{c}
 \\
\mathbf{x}_{1*} \\
\vdots \\
\mathbf{x}_{m*} \\
\mathbf{x}_{m+1*}
\end{array}
\overset{\mathbf{x}_{1*}\cdots\mathbf{x}_{m*}\quad \mathbf{x}_{m+1*}}{\left(
\begin{array}{c|c}
\mathbf{K}_{m*} & \mathbf{k}_{m+1*} \\
\hline
\mathbf{k}_{m+1*}^\mathsf{T} & k_{m+1*}
\end{array}
\right)} + \sigma^2 \mathbf{I}.
\tag{6}
$$

Here, we block partition the Cholesky factor $\mathbf{L}_{m+1}$ into three regions in the same manner:

$$
\mathbf{L}_{m+1} = 
\begin{array}{c}
1 \\
\vdots \\
m \\
m+1
\end{array}
\overset{1\cdots m \quad m+1}{\left(
\begin{array}{c|c}
\mathbf{L}_{11} & \mathbf{0} \\
\hline
\mathbf{l}_{21}^\mathsf{T} & l_{22}
\end{array}
\right)}.
\tag{7}
$$

Using the Cholesky factor update relations from Osborne (2010), $\mathbf{L}_{m+1}$ can be efficiently obtained using $\mathbf{L}_m$ as follows:

$$
\mathbf{L}_{11} = \mathbf{L}_m, \tag{8}
$$
$$
\mathbf{l}_{21} = \mathbf{L}_m \backslash \mathbf{k}_{m+1*}, \tag{9}
$$
$$
l_{22} = \sqrt{k_{m+1*} + \sigma^2 - \mathbf{l}_{21}^\mathsf{T} \mathbf{l}_{21}}. \tag{10}
$$

Here, $\mathbf{l}_{21} = \mathbf{L}_m \backslash \mathbf{k}_{m+1*}$ is obtained by solving the equation $\mathbf{L}_m \mathbf{l}_{21} = \mathbf{k}_{m+1*}$ for $\mathbf{l}_{21}$. This computation can be efficiently performed in $O(m^2)$ using back-substitutions Seeger (2004). The computational cost of the Cholesky factor update (equations 8, 9, 10) is dominated by equation 9, and as a result, $\mathbf{L}_{m+1}$ can be computed from $\mathbf{L}_m$ in $O(m^2)$. Once $\mathbf{L}_{m+1}$ is determined, the mean term (equation 4) $(\mathbf{L}_{m+1}\mathbf{L}_{m+1}^\mathsf{T})^{-1}\mathbf{y}^{m+1*}$ can also be computed in $O(m^2)$ by performing back-substitution twice.

## 4.2 COMPUTING INVERSE OF COVARIANCE MATRIX $\boldsymbol{\Sigma}_{m+1}^{-1}$

Once the Cholesky factor $\boldsymbol{\Sigma}_{m+1} = \mathbf{V}_{m+1}\mathbf{V}_{m+1}^\mathsf{T}$ is obtained, the first term in equation 3 can be computed by back-substitution in $O(N^2)$, and the second term, which is the product of the diagonal components of the Cholesky factor, can be computed in $O(N)$. Here, we describe a method to efficiently compute $\mathbf{V}_{m+1}$ using $\mathbf{V}_m$.

The covariance matrix can be transformed as follows:

$$
\mathbf{V}_{m+1}\mathbf{V}_{m+1}^\mathsf{T} = [\mathbf{K}_N + \sigma^2\mathbf{I}] - \mathbf{K}_{N,m+1*}^\mathsf{T}[\mathbf{K}_{m+1*} + \sigma^2\mathbf{I}]^{-1}\mathbf{K}_{N,m+1*} \tag{11}
$$
$$
= [\mathbf{K}_N + \sigma^2\mathbf{I}] - \mathbf{K}_{N,m+1*}^\mathsf{T}\left(\mathbf{L}_{m+1}\mathbf{L}_{m+1}^\mathsf{T}\right)^{-1}\mathbf{K}_{N,m+1*} \tag{12}
$$
$$
= [\mathbf{K}_N + \sigma^2\mathbf{I}] - \left(\mathbf{L}_{m+1}^{-1}\mathbf{K}_{N,m+1*}\right)^\mathsf{T}\left(\mathbf{L}_{m+1}^{-1}\mathbf{K}_{N,m+1*}\right). \tag{13}
$$

Here, if we let $\mathbf{L}_{m+1}^{-1}\mathbf{K}_{N,m+1*} = \mathbf{A}_{m+1}$, we want to show that it actually has the structure:

$$
\mathbf{A}_{m+1} = \left(
\begin{array}{c}
\mathbf{A}_m \\
\hline
\mathbf{a}_{m+1}^\mathsf{T}
\end{array}
\right).
\tag{14}
$$

Upon rearranging terms, it can be written as $\mathbf{L}_{m+1}\mathbf{A}_{m+1} = \mathbf{K}_{N,m+1*}$. The explicit structures of $\mathbf{L}_{m+1}$ and $\mathbf{K}_{N,m+1*}$ can be represented as follows:

$$
\left(
\begin{array}{c}
\mathbf{L}_m \\
\hline
\mathbf{l}_{m+1}^\mathsf{T}
\end{array}
\right)\mathbf{A}_{m+1} = \left(
\begin{array}{c}
\mathbf{K}_{N,m*} \\
\hline
\mathbf{k}_{N,m+1}^\mathsf{T}
\end{array}
\right).
\tag{15}
$$

When solving this for each column of $\mathbf{A}_{m+1}$ using the back-substitution method, it is observed that the components of each column are only influenced by the components above them in $\mathbf{L}_{m+1}$ and $\mathbf{K}_{N,m+1*}$. Therefore, the influences of $\mathbf{l}_{m+1}^{\mathsf{T}}$ and $\mathbf{k}_{N,m+1}^{\mathsf{T}}$ are limited to the final row, resulting in the form described in Equation 14. Here, the $n$th component of $\mathbf{a}_{m+1}^{\mathsf{T}}$, $a_{m+1,n}$, is defined as follows. Let $\mathbf{l}_{m+1}^{\mathsf{T}} = (\mathbf{l}_{m+1,\leq m}^{\mathsf{T}} | l_{m+1,m+1})$, the vector excluding the last element $a_{m+1,n}$ of the $n$th column of $\mathbf{A}_{m+1}$ be $\mathbf{a}_{m,n}$, and the $n$th component of $\mathbf{k}_{N,m+1}^{\mathsf{T}}$ be $k_{m+1,n}$. Then,

$$a_{m+1,n} = \frac{1}{l_{m+1,m+1}} \left\{ k_{m+1,n} - (\mathbf{l}_{m+1,\leq m}^{\mathsf{T}} \mathbf{a}_{m,n}) \right\} \tag{16}$$

can be obtained.

The second term of Equation 13 is expressed as $\mathbf{A}_{m+1}^{\mathsf{T}}\mathbf{A}_{m+1}$, but by leveraging the structure of Equation 14, it can be separated into $\mathbf{A}_{m+1}^{\mathsf{T}}\mathbf{A}_{m+1} = \mathbf{A}_m^{\mathsf{T}}\mathbf{A}_m + \mathbf{a}_{m+1}\mathbf{a}_{m+1}^{\mathsf{T}}$. Substituting this in, we get

$$\mathbf{V}_{m+1}\mathbf{V}_{m+1}^{\mathsf{T}} = [\mathbf{K}_N + \sigma^2\mathbf{I}] - \left( \mathbf{A}_m^{\mathsf{T}}\mathbf{A}_m + \mathbf{a}_{m+1}\mathbf{a}_{m+1}^{\mathsf{T}} \right). \tag{17}$$

From Equation 13, it is known that $[\mathbf{K}_N + \sigma^2\mathbf{I}] - \left( \mathbf{A}_m^{\mathsf{T}}\mathbf{A}_m \right) = \mathbf{V}_m\mathbf{V}_m^{\mathsf{T}}$, hence

$$\mathbf{V}_{m+1}\mathbf{V}_{m+1}^{\mathsf{T}} = \mathbf{V}_m\mathbf{V}_m^{\mathsf{T}} - \mathbf{a}_{m+1}\mathbf{a}_{m+1}^{\mathsf{T}} \tag{18}$$

holds true. With this form, the rank-one downdate Seeger (2004) can be utilized to update from $\mathbf{V}_m$ to $\mathbf{V}_{m+1}$ with a computational complexity of $O(N^2)$. The derived algorithm is summarized in Algorithm 1 in Appendix B.

## 5 EXPERIMENTS

### 5.1 COMPARISON WITH OTHER METHODS

Firstly, in order to measure the accuracy of the proposed method in regression, we compare the mean squared error (MSE) with existing methods. For the first dataset, we employ one-dimensional generated data to investigate the behavior under the simplest scenario. We aim to replicate a scenario where we can obtain training data from the true distribution and data from a partially correct simulator. In this experiment, the distributions of the true and the simulator are given by the following equations:

$$\begin{aligned} True \quad &: \quad 4\cos(1.5x)\exp(-0.1x) + 4\arctan(x-10) + \mathcal{N}(0, 0.5), \\ Sim. \quad &: \quad 4\cos(1.5x) + 4\arctan(x-10) + \mathcal{N}(0, 0.5). \end{aligned}$$

It is assumed that during the simulator's construction, the existence of the decay term $\exp(-0.1x)$ was not recognized. The proposed algorithm aims to selectively incorporate only the matching data from simulator data into the training data. In our experiments, we utilize three types of data: training data, simulation data, and test data. To ensure these data sets do not overlap, they were constructed as follows: From the true distribution, 2000 data points were generated, and randomly $50, 100, 200, \ldots, 900$ points were chosen as training data. From the remaining data, 1000 data points were randomly selected as test data. As simulator data, 10000 data points were generated from the simulation distribution. In most of our experiments, we repeated the experiment 10 times, reporting the average and standard deviation. The training and test data were reselected randomly from the 2000 data points generated from the true distribution in each trial.

For the second dataset, we utilize eight-dimensional generated data to evaluate the effectiveness of the proposed method when extended to multidimensional settings. The distributions for the true and the simulator are as follows:

$$\begin{aligned} True \quad &: \quad 4\cos 1.5x_1 + 3\sin 0.5x_2 + 4x_3 + (x_4 - 5)^2 + 3\cos 3x_5 + 3\sin x_6 + x_7^2 - (x_8 - 15)^2 \\ &\quad + 150\exp\left(-(x_1 - 10)^2\right) + \mathcal{N}(0, 1.0), \\ Sim. \quad &: \quad 4\cos 1.5x_1 + 3\sin 0.5x_2 + 4x_3 + (x_4 - 5)^2 + 3\cos 3x_5 + 3\sin x_6 + x_7^2 - (x_8 - 15)^2 \\ &\quad + \mathcal{N}(0, 1.0). \end{aligned}$$

The methods for generating training data, simulation data, and test data are the same as for the one-dimensional data mentioned above. However, to reduce learning time in the eight-dimensional data, we generated and used 1000 data points as simulation data. For details on the kernel and training parameters, refer to Appendix C.

For the third dataset, we use a publicly available dataset to validate the performance on real-world data. We evaluated our method using the concrete compressive strength dataset, a standard dataset in regression tasks Yeh (2007). The concrete compressive strength dataset aims to predict the target variable, concrete compressive strength, using eight explanatory variables related to materials. As for data generated from a simulator, out of 500 data points, the target variable was modified to be $+20$ from its original value for $3/4$ of the data, while the remaining $1/4$ were kept as the original data. This simulates a scenario where a simulator tends to overestimate the concrete compressive strength due to some factors. The number of training data points was set at $100, 200, 300, 400$, and the test data comprised 100 data points. For details on the kernel and training parameters, refer to Appendix D.

For the fourth dataset, we leverage an off-the-shelf simulator, as mentioned in the introduction, to assess the proposed method in realistic situations. Suppose we want to estimate the power generation for the next year's weather conditions based on hourly weather and power generation data from the past year and a power generation simulator. However, we assume a situation where the inverter settings in the power generation simulator are incorrect. As an off-the-shelf simulator, we used pvlib python Anderson et al. (2023), a Python simulator for photovoltaic energy systems. In our experiments, the training and test data were generated using the Sandia Inverter, while the generated data from the power generation simulator was obtained using the CEC Inverter. The weather data used was from Albuquerque for the years (2005, 2006, 2008, 2009, 2011, 2013, 2014, 2015)[5]. For each dataset - training, test, and simulator-generated - we used one year's worth of data ($24 \times 365$ data points). For example, the training and simulator data were from 2005, while the test data was from the following year, 2006. We evaluated the performance using seven such combinations.

As comparative methods, in addition to the standard GPs, we implemented a method using anomaly detection as a naive benchmark and SoD Lalchand & Faul (2018). Lalchand & Faul (2018) described in Appendix A.1, promote diversity of training data. The termination condition was set to accept until half of all simulator data were accepted. The method using anomaly detection is a naive approach that excludes outliers in the simulator using a GP trained on true training data. The criterion for determining if it's an outlier is calculated by feeding all input data $\mathbf{X}$ from the simulator to the GP to predict its distribution and then using the likelihood of all the output data $\mathbf{y}$ from the simulator. If the likelihood is below a certain threshold, it is considered an outlier and is not added to the training data. The threshold was selected from $(0.2, 0.4, 0.6, 0.8, 1.0)$ through cross-validation. This method can be seen as not sequentially adding simulator data to training data in the proposed method, and also measures the significance of sequential addition.

Figure 1 (a) to (c) shows the results of comparing the MSE on three datasets. The horizontal axis of each graph represents the number of true training data used for learning, while the vertical axis represents MSE on test data, with a lower value being better. The solid lines represent the average of 10 trials for each method, and the shaded regions represent the standard deviation. Figure 1 (d) presents the MSE of each method on the pvlib python dataset. The results shown are the average values over the seven years of data.

For those datasets, improvements were observed with the proposed method, negative marginal likelihood (NML), compared to the Plane GP. As can be seen in Section 5.2, this is because only beneficial data from the simulator-generated data were added to the training data.

## 5.2 PROPERTIES OF MARGINAL LIKELIHOOD AND SOD

We aim to clarify the effective range of marginal likelihood and SoD. Specifically, unlike our proposed method which doesn't require training data to be from the true distribution, SoD assumes it is. We seek to understand the effects on both methods when training data diverges from the true distribution.

---

[5]Although the years are intermittent, these eight years of data are all the data included in the library

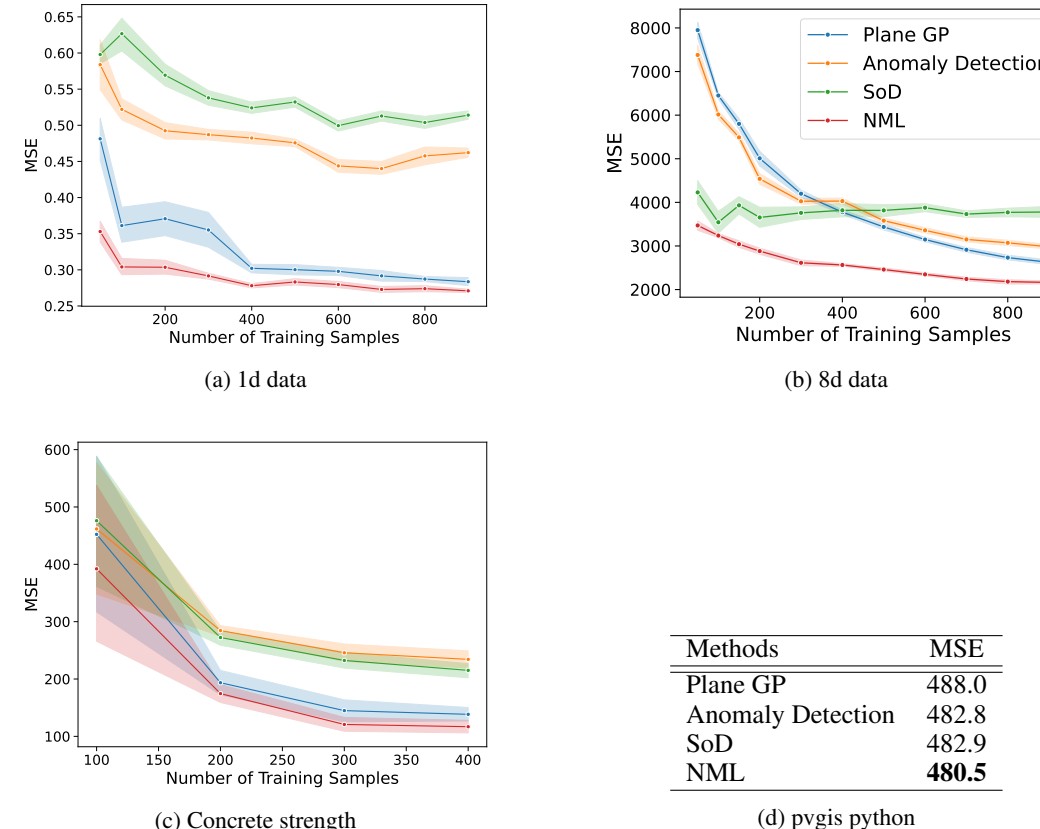

Figure 1: Comparison of the proposed method (NML) with other methods

To this end, we altered the distance between the simulator distribution and the true distribution, observing the prediction accuracy of each method. Specifically, the true and simulator distributions were set as follows:

$$\text{True} \quad : \quad \sin(x) + \arctan(x - 10) + \mathcal{N}(0, 0.5)$$
$$\text{Sim.} \quad : \quad a \times \sin(x) + \arctan(x - 10) + \mathcal{N}(0, 0.5)$$

Here, $a$ is an experimental parameter to adjust the distance between the simulator and true distributions. In this experiment, $a = [1, 2, 3]$. When $a = 1$, the training data candidates are assumed to be sampled from the true distribution, and as $a$ increases, the distribution of training data candidates diverges from the true distribution. The number of true training data was set to 50, the number of training data candidates generated from the simulator was 1000, and the number of test data was 1,000. We resampled all this data 10 times and showed the average and standard deviation of MSE on a graph. For details on the kernel and training parameters, refer to Appendix D.

The results are shown in Figure 2. First, when $a = 1$ and the distributions of training data candidates and the true distribution match, both NML and SoD had a lower MSE than Plane GP. Among them, SoD had a lower MSE than NML. This suggests that the diversity of training data in the SoD approach is important under this condition. However, as $a$ increases to 2 and 3, and the training data candidate distribution deviates from the true distribution, the MSE of SoD increases, while the MSE of NML remains smaller than Plane GP. For $a = 3$, data included in the training data from the training data candidates and data that was rejected are visualized in Figure 3.

As SoD aims to enhance the diversity of training data, it accepts training data candidates that deviate from the true distribution that training and test data follow, resulting in a deterioration in MSE due to the predictions being influenced by these data. On the other hand, NML mainly accepts training data candidates close to the true distribution, leveraging the correct part of the simulator distribution, $arctan(x - 10)$. Accepting only the beneficial parts and stopping there implies that we have

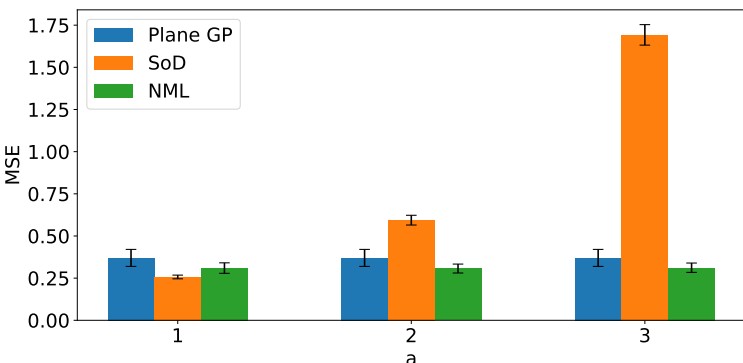

Figure 2: Impact of the difference between the training data candidate distribution and the true distribution on the MSE of each method.

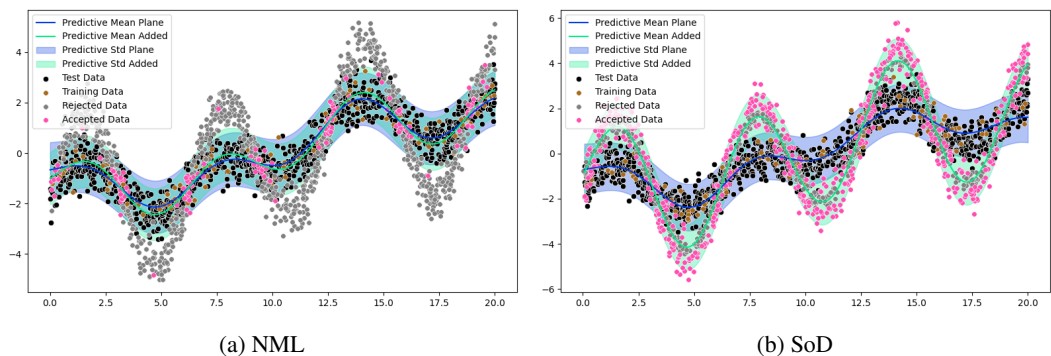

(a) NML

(b) SoD

Figure 3: Data adopted for training from the training data candidates. The x-axis represents $x$ and the y-axis represents $y$. Predictive Mean Plane and Predictive Std Plane are the average and standard deviation of Plane GP predictions. Predictive Mean Added and Predictive Std Added are the average and standard deviation of predictions for NML or SoD. Accepted Data are those adopted from the training data candidates, and the gray points were not adopted.

achieved our original goal, which was to automatically adjust the strength of the model's assumptions, that is, to find an appropriate balance between the strong assumptions of the simulator and the lax assumptions of the GPs. From these results, when training data deviates from the true distribution and aligns with simulator-generated data, using marginal likelihood to fit training data can yield samples close to the true distribution, benefiting prediction tasks. In contrast, the SoD metric, focusing on training data diversity, can include data far from the true distribution in simulator-generated data, negatively impacting predictions.

## 5.3 COMPUTATION TIME

We experimentally verify the computational complexity reduction effect of the proposed method. We compare the method that goes through the Cholesky decomposition of the covariance matrix proposed in Section 4 $\Sigma_{m+1} = \mathbf{V}_{m+1}\mathbf{V}_{m+1}^\top$ with the method that does not. The CPU was an Intel CORE i7 vPro 8th Gen, and 32GB memory was used. The results are shown in Figure 4. The horizontal axis represents the number of training data $N$, and the vertical axis represents the time taken for learning in hours. With the proposed computational technique, the increase in computational time is gradual even as the

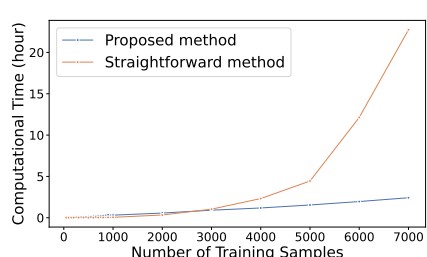

Figure 4: Computational Time

number of training data increases, while without it, there
is a sharp increase. This result confirms that the computa-
tional time order of the conventional method is $O(N^3)$, whereas the computational time order of the
proposed method is $O(N^2)$. Furthermore, the gentle increase in computational time of the proposed
method suggests that the coefficient of $O(N^2)$ is small.

## 6 CONCLUSION

We propose using the negative log marginal likelihood of the GPs as a criterion when selectively
adding simulator-generated data to the training data. Through experiments, it was confirmed that
the proposed method extracts only the correct knowledge from the simulator and improves the MSE.
Moreover, by taking into account the Cholesky factor and the dependency of matrix elements, We
proposed an algorithm that reduces the computational cost of selecting training data candidates from
$O(M^3N + M^2N^2 + MN^3)$ to $O(M^2N + MN^2)$.

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

Methods in Science and Engineering*, 2018.

Neil Lawrence, Matthias Seeger, and Ralf Herbrich. Fast sparse gaussian process methods: the
informative vector machine. In *Neural Information Processing Systems (NeurIPS)*, 2002.

Sungbin Lim, Ildoo Kim, Taesup Kim, Chiheon Kim, and Sungwoong Kim. Fast autoaugment. In
*Neural Information Processing Systems (NeurIPS)*, 2019.

Duy Nguyen-Tuong, Jan Peters, and Matthias Seeger. Local gaussian process regression for real
time online model learning and control. In *Neural Information Processing Systems (NeurIPS)*,
2008.

Michael A Osborne. *Bayesian Gaussian processes for sequential prediction, optimisation and quadrature*. PhD thesis, Oxford University, UK, 2010.

Barnabás Póczos and Jeff Schneider. Nonparametric estimation of conditional information and divergences. In *Artificial Intelligence and Statistics*, 2012.

Maziar Raissi, Paris Perdikaris, and George E Karniadakis. Physics-informed neural networks: A deep learning framework for solving forward and inverse problems involving nonlinear partial differential equations. *Journal of Computational physics*, 2019.

Matthias Seeger. Low rank updates for the cholesky decomposition. Technical report, 2004.

Matthias W Seeger, Christopher KI Williams, and Neil D Lawrence. Fast forward selection to speed up sparse gaussian process regression. In *International Workshop on Artificial Intelligence and Statistics (AISTATS)*, 2003.

Jonathon Shlens. Notes on kullback-leibler divergence and likelihood theory. *Systems Neurobiology Laboratory*, 92037:1–4, 2007.

Edward Snelson and Zoubin Ghahramani. Sparse gaussian processes using pseudo-inputs. In *Neural Information Processing Systems (NeurIPS)*, 2005.

Michalis Titsias. Variational learning of inducing variables in sparse gaussian processes. In *Artificial intelligence and statistics*, 2009.

I-Cheng Yeh. Concrete Compressive Strength. UCI Machine Learning Repository, 2007.

# A EXTENDED RELATED WORKS

## A.1 SUBSET OF DATA

Subset of Data (SoD) is a method that selects important data from candidate training data for GPs. By generating training data candidates from simulators built using domain knowledge, the extent of domain knowledge reflection into GPs can be adjusted by determining which data and how much of it is included in the training data. SoD evaluates the training data candidates based on a metric and greedily adds the highest-ranked ones. The metric for selecting important data is the diversity of the training data Seeger et al. (2003); Lawrence et al. (2002); Lalchand & Faul (2018). Various methods to measure this diversity have been proposed. Lawrence et al. (2002) uses the difference in entropy between predictive distributions with and without a specific training data point as its metric. Seeger et al. (2003) uses the KL divergence between predictive distributions with and without a specific training data point. Lalchand & Faul (2018) employs its unique diversity metric. Specifically, when predicting with a GPs using the already accepted training data, they add data candidates to the training set where the sum of the squared error and the prediction uncertainty is large. As the training data candidates move farther from the already adopted training data, both the prediction squared error and uncertainty increase. Thus, candidates that enhance the diversity of the training data are chosen. These methods assume that training data candidates are sampled from the true distribution and can ensure diversity within the training set. However, in the context of adjusting the amount of domain knowledge introduced, training data candidates are sampled from simulators using domain knowledge, which means there could be regions that deviate from the true distribution. Consequently, data deviating from the true distribution might be prioritized, causing predictive distributions to diverge. Lawrence et al. (2002) employs a method using GPs classification, while Seeger et al. (2003) uses a sparse GPs method. Given this, we choose Lalchand & Faul (2018), which can be directly applied to standard GPs regression, to represent the SoD method and compare it with our proposed method in experiments.

## A.2 SPARSE GAUSSIAN PROCESSES

Similarly to the Subset of Data, there are methods to generate a small number of pseudo-training data for reducing the training data. Some of these methods, like the proposed approach, use the log marginal likelihood as a metric to generate pseudo-training data Titsias (2009);

Snelson & Ghahramani (2005). However, none of these compute the exact log marginal likelihood. For example, Titsias (2009) developed sparse GPs to reduce training data by substituting the Gaussian process model from $p(\mathbf{f}|\mathbf{X})$ to $p(\mathbf{f}|\mathbf{f}^m)$, avoiding direct dependency of function values $\mathbf{f}$ on the training data $\mathbf{X}$, rather relying of function value $\mathbf{f}^m$ on the pseudo data. This change in the model altered the formula for marginal likelihood, which could not be computed quickly, leading to the use of a lower bound of marginal likelihood as the metric instead Titsias (2009). On the other hand, in our intended applications, all training data are available, so there is no need to alter the marginal likelihood. Our proposed acceleration method allows us to use the exact log likelihood as the metric. Similarly, Snelson & Ghahramani (2005) also uses an approximation of the marginal log likelihood, not the exact value, as the metric Titsias (2009).

### A.3  RECOMPUTATION TECHNIQUES FOR GPS WHEN DATA IS ADDED

SoD evaluates the goodness of added training data candidates by sequentially adding them to the GP's training data and measuring the predictive distribution of the model using a metric. The computational effort needed to compute the predictive distribution when data is added has been researched in the domain of Online GPs. After training with $N$ data points (after computing the inverse matrix with $N$ data), an additional data point can be learned with an added computational effort of $O(N^2)$ using a technique called rank-one update Nguyen-Tuong et al. (2008); Seeger (2004). This technique is used in part in our method. However, even with this technique, it's not efficient to compute the inverse of the covariance matrix for the log marginal likelihood. In this study, we propose a method to efficiently compute it.

Since it takes $O(N^3)$ computational effort to compute the predictive distribution of a GPs, methods have been proposed that combine approximation techniques and online learning to reduce this. Among the approximation methods that use the inducing variable method and variational inference, methods to go online by mini-batch Hensman et al. (2013); Cheng & Boots (2016) and methods to go online by sequential Bayesian updates Csató & Opper (2002); Bui et al. (2017) have been proposed. Also, an online method that used a local GPs has been proposed Nguyen-Tuong et al. (2008). Although our study does not use these approximation methods to compute the log marginal likelihood without approximation, they may be utilized in the future to reduce computational effort.

### A.4  AUTO-DATAAUGMENTATION

When Data Augmentation rules are considered as one of the inductive biases, one can interpret it as injecting domain knowledge, such as the rules of augmentation (like an image retains its class even when flipped), into the machine learning model. Notably, auto data augmentation Cubuk et al. (2019; 2020); Ho et al. (2019); Lim et al. (2019) explores optimal magnitudes and application probabilities of multiple data augmentations, like image flips and rotations, based on training data. This concept is similar to our proposed method. The difference lies in the fact that while auto data augmentation employs neural network classifiers as the discriminative model, our approach uses a GPs regression model. Moreover, auto data augmentation evaluates policies, while our method evaluates individual single data. Although potentially less efficient, our method offers versatility in situations where defining a policy is challenging. Our approach does not optimize the order of samples for evaluation. This is because, compared to the vast neural networks used in auto data augmentation, the GPs evaluations can be conducted in a shorter computation time. However, considering sample order optimization could be a consideration for future work if it becomes crucial.

## B  DETAILS OF THE PROPOSED ALGORITHM

The pseudo-code of the proposed algorithm is shown in Algorithm 1.

The hyperparameters of the input kernel function are pre-optimized using the true training data $(\mathbf{X}^N, \mathbf{y}^N)$ and remain fixed until the completion of Algorithm 1. While it is possible to relearn the kernel's hyperparameters using the selected generated data $(\mathbf{X}^{M^*}, \mathbf{y}^{M^*})$ and the original training data $(\mathbf{X}^N, \mathbf{y}^N)$ after applying Algorithm 1, this second round of learning has not been conducted in the experiments presented in this paper.

---

**Algorithm 1** Selection of Samples Reducing Free Energy

---

**Input:** Training data $(\mathbf{X}^N, \mathbf{y}^N)$, simulator $p_S(y, \mathbf{x})$, GP kernel function
**Output:** Selected generated data $\left(\mathbf{X}^{M^*}, \mathbf{y}^{M^*}\right)$

 1: compute $\mathbf{K}_N$
 2: $\mathbf{V}_0 = Cholesky\left(\mathbf{K}_N + \sigma^2 \mathbf{I}\right)$
 3: **while** True **do**
 4:     sample $(y_{m+1^*}, \mathbf{x}_{m+1^*}) \sim p_S(y, \mathbf{x})$
 5:     **if** $m = 0$ **then**
 6:         $\mathbf{L}_1 = \sqrt{k_{1^*} + \sigma^2}$
 7:     **else** $\{m \geq 1\}$
 8:         compute $\mathbf{k}_{m+1^*}, k_{m+1^*}$
 9:         $\mathbf{L}_{m+1} = CholeskyFactorUpdate\left(\mathbf{L}_m, \mathbf{k}_{m+1^*}, k_{m+1^*} + \sigma^2\right)$
10:     **end if**
11:     compute $\mathbf{K}_{N,m+1^*}$
12:     Mean $\boldsymbol{\mu}_{m+1} = \mathbf{K}_{N,m+1^*}^{\mathsf{T}}\left(\mathbf{L}_{m+1}\mathbf{L}_{m+1}^{\mathsf{T}}\right)^{-1}\mathbf{y}^{m+1}$
13:     **if** $m = 0$ **then**
14:         $\mathbf{a}_1^{\mathsf{T}} = \mathbf{K}_{N,1^*}/\mathbf{L}_1$
15:         $\mathbf{A}_1 = \mathbf{a}_1^{\mathsf{T}}$
16:     **else** $\{m \geq 1\}$
17:         $\mathbf{a}_{m+1} = LastCholeskySolution\left(\mathbf{L}_{m+1}, \mathbf{A}_m, \mathbf{K}_{N,m+1^*}\right)$
18:         $\mathbf{A}_{m+1} = stack\left(\mathbf{A}_m, \mathbf{a}_{m+1}\right)$
19:     **end if**
20:     $\mathbf{V}_{m+1} = RankOneDowndate\left(\mathbf{V}_m, \mathbf{a}_{m+1}\right)$
21:     $\left(\mathbf{y}^N - \boldsymbol{\mu}_{m+1}\right)^{\mathsf{T}}\boldsymbol{\Sigma}_{m+1}^{-1}\left(\mathbf{y}^N - \boldsymbol{\mu}_{m+1}\right) = \left(\mathbf{y}^N - \boldsymbol{\mu}_{m+1}\right)^{\mathsf{T}}\left(\mathbf{V}_{m+1}\mathbf{V}_{m+1}^{\mathsf{T}}\right)^{-1}\left(\mathbf{y}^N - \boldsymbol{\mu}_{m+1}\right)$

22:     $|\boldsymbol{\Sigma}_{m+1}| = $ product of diagonal elements of $\mathbf{V}_{m+1}$
23:     $F_{m+1} = -\frac{1}{2}\left(\mathbf{y}^N - \boldsymbol{\mu}_{m+1}\right)^{\mathsf{T}}\boldsymbol{\Sigma}_{m+1}^{-1}\left(\mathbf{y}^N - \boldsymbol{\mu}_{m+1}\right) - \frac{1}{2}\log|\boldsymbol{\Sigma}_{m+1}| - \frac{N}{2}\log 2\pi$
24:     **if** $F_{m+1} < F_m$ **then**
25:         $\mathbf{X}^{m+1} = \mathbf{X}^{m^*} + \mathbf{x}_{m+1^*}$
26:         $\mathbf{y}^{m+1} = \mathbf{y}^{m^*} + y_{m+1^*}$
27:         $m{+}{+}$
28:     **else** $\{F_{m+1} \geq F_m\}$
29:         reject $(y_{m+1^*}, \mathbf{x}_{m+1^*})$
30:         break if rejected $R$ times consecutively
31:     **end if**
32: **end while**

---

## C  DETAILS OF KERNELS AND TRAINING PARAMETERS IN SECTION 5.1

We employed the RBF kernel as the kernel for GPs regression. The initial values of the kernel parameters were set to $amplitude = 10$ and $length_s cale = 10$. The variance of observation noise was 1.

The estimation of the above three parameters was carried out by maximum likelihood estimation using the true training data. For training parameters, we used Adam Kingma & Ba (2015) with a learning rate of $0.001$, $\beta_1 = 0.9$, $\beta_2 = 0.999$, and $\epsilon = 1e - 7$. The batch size was set to 1, and the number of iterations was 15,000. We did not optimize these training hyperparameters.

## D  DETAILS OF KERNELS AND TRAINING PARAMETERS IN SECTION 5.2 AND REAL DATA

We employed the RBF kernel as the kernel for GPs regression. The initial values of the kernel parameters were set to $amplitude = 1$ and $length\_scale = 1$. The variance of observation noise was 1. Estimation of these parameters was conducted using maximum likelihood estimation with true training data. For training parameters, we used Adam Kingma & Ba (2015) with a learning rate

of 0.01, $\beta_1 = 0.9$, $\beta_2 = 0.999$, and $\epsilon = 1e - 7$. The batch size was set to 1, and the number of iterations was 2,000. These training hyperparameters were not optimized.

# E  ABLATION STUDY

Our proposed method used the free energy as a metric, and efficiently computed the computationally expensive inverse of the covariance matrix $\mathbf{\Sigma}_{m+1}^{-1}$ and its determinant $\|\mathbf{\Sigma}_{m+1}\|$ in the free energy equation 3. To verify whether it's truly necessary to compute the inverse and determinant of the covariance matrix, we conducted an ablation study.

We compared with two ablated methods. The first method removed the inverse of the covariance matrix from equation 3, resulting in $-\frac{1}{2}(\mathbf{y}^N - \boldsymbol{\mu}_{m+1})^\mathsf{T}(\mathbf{y}^N - \boldsymbol{\mu}_{m+1}) - \frac{1}{2}\log|\mathbf{\Sigma}_{m+1}| - \frac{N}{2}\log 2\pi$. The second method also eliminated the determinant term, yielding $-\frac{1}{2}(\mathbf{y}^N - \boldsymbol{\mu}_{m+1})^\mathsf{T}(\mathbf{y}^N - \boldsymbol{\mu}_{m+1}) - \frac{N}{2}\log 2\pi$. This approach is simply the squared error between the GPs (GP) prediction mean and the actual data. The constant term $\frac{N}{2}\log 2\pi$ does not influence data selection.

We varied the amount of training data, conducted a 10-cross validation, and compared the MSE of test data. The results are shown in Figure 5. The method without the inverse of the covariance matrix is labeled as w/o inverse, the one without the determinant is labeled as squared error, and our proposed method is labeled as NML. The results showed that MSE, which evaluates the full free

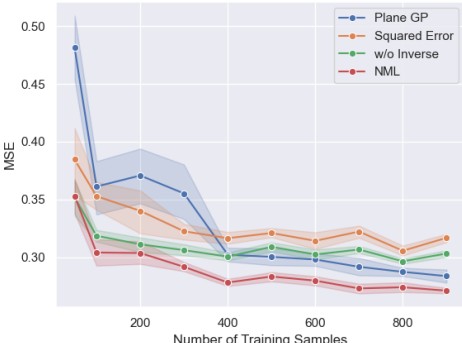

Figure 5: Comparison of MSE with ablated methods

energy, demonstrated more stable improvements than both the simple squared error and w/o inverse. From this, we can deduce that both the inverse of the covariance matrix $\mathbf{\Sigma}_{m+1}^{-1}$ and its determinant $\|\mathbf{\Sigma}_{m+1}\|$ significantly contribute to the improvement of MSE.

# F  NEGATIVE LOG MARGINAL LIKELIHOOD

In Bayesian statistics, free energy (negative log marginal likelihood) or generalization loss is used as a model evaluation metric. The free energy measures the model's fit to the training data, and model selection often involves either the free energy or its approximation, the BIC (Bayes Information criterion). The generalization loss measures the accuracy of the model's predictive distribution, and for model selection, cross-validation loss or the AIC (Akaike Information Criterion) Akaike (1998) are commonly used. In this study, we use the free energy as the metric for SoD.

We define the symbols as follows: $\mathbf{X} \in \mathbb{R}^d$ is a random variable, $\mathbf{X}^N = (\mathbf{X}_1, \mathbf{X}_2, \ldots, \mathbf{X}_N)$ are $N$ independent random variables following the same distribution, $\mathbf{y} \in \mathbb{R}^1$ is a random variable, and $\mathbf{y}^N = (\mathbf{y}_1, \mathbf{y}_2, \ldots, \mathbf{y}_N)$ are $N$ independent random variables following the same distribution. The free energy of the discriminative model (including GPs) is given by the following equation:

$$F = -\log p(\mathbf{y}^N | \mathbf{X}^N). \tag{19}$$

When the realized values of $N$ training data are obtained in a regression task, the realized value of the model's free energy is $-\log p(\mathbf{y}^N = \mathbf{y}^N | \mathbf{X}^N = \mathbf{x}^N)$. Random variables are denoted in uppercase, and realized values are denoted in lowercase.

This metric is explained by the difference between the true distribution of the dataset and the inferred model distribution. If we denote the true distribution of the dataset as $q(\mathbf{y}^N | \mathbf{X}^N) q(\mathbf{X}^N)$ and the distribution in the discriminative model of the dataset as $p(\mathbf{y}^N | \mathbf{X}^N)$, the conditional KL-divergence Póczos & Schneider (2012) between the two distributions is

$$KL(q(\mathbf{y}^N | \mathbf{X}^N) || p(\mathbf{y}^N | \mathbf{X}^N)) = \int q(\mathbf{X}^N) \int q(\mathbf{y}^N | \mathbf{X}^N) \log \frac{q(\mathbf{y}^N | \mathbf{X}^N)}{p(\mathbf{y}^N | \mathbf{X}^N)} d\mathbf{y}^N d\mathbf{X}^N \quad (20)$$
$$= \mathbb{E}[F] + C.$$

Where $\mathbb{E}[\cdot]$ denotes the average over samples from the true distribution and $C = \int q(\mathbf{y}^N | \mathbf{X}^N) q(\mathbf{X}^N) \log q(\mathbf{y}^N | \mathbf{X}^N) d\mathbf{X}^N d\mathbf{y}^N$ is a constant that does not depend on the model's distribution. Therefore, a smaller free energy $F$ indicates that the inferred distribution approximates the true distribution well on average.

## G  FREE ENERGY WHEN GENERATED DATA IS ADDED IN GENERAL

As mentioned at the beginning of Section 3, we want to measure whether the performance of the discriminative model improved by adding samples from the simulator to the training data of the discriminative model. We adopt free energy as a performance metric and extend the free energy of Equation 19 when samples are added from the simulator. If $m$ samples from the simulator are represented by $(\mathbf{X}^{m^*}, \mathbf{y}^{m^*})$, then the predictive distribution given $(\mathbf{X}^{m^*}, \mathbf{y}^{m^*})$ in the model becomes $p(\mathbf{y}^N | \mathbf{X}^N, \mathbf{X}^{m^*}, \mathbf{y}^{m^*})$. The conditional KL-divergence Póczos & Schneider (2012) between $p(\mathbf{y}^N | \mathbf{X}^N, \mathbf{X}^{m^*}, \mathbf{y}^{m^*})$ and the true distribution can be transformed as follows:

$$KL(q(\mathbf{y}^N | \mathbf{X}^N) \quad || \quad p(\mathbf{y}^N | \mathbf{X}^N, \mathbf{X}^{m^*}, \mathbf{y}^{m^*})) \quad (21)$$
$$= \int q(\mathbf{X}^N) \int q(\mathbf{y}^N | \mathbf{X}^N) \log \frac{q(\mathbf{y}^N | \mathbf{X}^N)}{p(\mathbf{y}^N | \mathbf{X}^N, \mathbf{X}^{m^*}, \mathbf{y}^{m^*})} d\mathbf{y}^N d\mathbf{X}^N$$
$$= \mathbb{E}[-\log p(\mathbf{y}^N | \mathbf{X}^N, \mathbf{X}^{m^*}, \mathbf{y}^{m^*})] + C.$$

Therefore, if we define

$$F_{m^*} = -\log p(\mathbf{y}^N | \mathbf{X}^N, \mathbf{X}^{m^*}, \mathbf{y}^{m^*}) \quad (22)$$

then minimizing $F_{m^*}$ will minimize $KL(q(\mathbf{y}^N | \mathbf{X}^N) || p(\mathbf{y}^N | \mathbf{X}^N, \mathbf{X}^{m^*}, \mathbf{y}^{m^*}))$ on average. Thus, we obtained $F_{m^*}$ as a performance metric for the discriminative model when $(\mathbf{X}^{m^*}, \mathbf{y}^{m^*})$ is given. $F_{m^*}$ is, then, the free energy when generated data is added.

## H  BASICS OF GPS REGRESSION

Given an input $\mathbf{x}$, we define the feature vector of $\mathbf{x}$ as $\phi(\mathbf{x}) = (\phi_0(\mathbf{x}), \phi_1(\mathbf{x}), \dots, \phi_H(\mathbf{x}))^\mathsf{T}$. Considering the linear regression model $y = \mathbf{w}^\mathsf{T} \phi(\mathbf{x})$ with weights $\mathbf{w} = (w_0, w_1, \dots, w_H)$, for $N$ input-output pairs, it can be described simultaneously using the design matrix $\mathbf{\Phi} = (\phi(\mathbf{x}_1)^\mathsf{T}, \dots, \phi(\mathbf{x}_N)^\mathsf{T})$, which can be written as $\mathbf{y} = \mathbf{\Phi} \mathbf{w}$. Here, $\mathbf{y} = (y_1, \dots, y_N)^\mathsf{T}$.

Assume the weights $\mathbf{w}$ are drawn from a Gaussian distribution $\mathcal{N}(\mathbf{0}, \lambda^2 \mathbf{I})$ with mean $\mathbf{0}$ and variance $\lambda \mathbf{I}$. Then, since $\mathbf{y}$ is a linear transformation of the Gaussian distributed vector $\mathbf{w}$ by the constant matrix $\mathbf{\Phi}$, $\mathbf{y} = \mathbf{\Phi} \mathbf{w}$ also follows a Gaussian distribution. The mean is given by $\mathbb{E}[\mathbf{y}] = \mathbb{E}[\mathbf{\Phi} \mathbf{w}] = \mathbf{\Phi} \mathbb{E}[\mathbf{w}] = \mathbf{0}$, and the covariance matrix is $\mathbb{E}[\mathbf{y} \mathbf{y}^\mathsf{T}] - \mathbb{E}[\mathbf{y}] \mathbb{E}[\mathbf{y}]^\mathsf{T} = \mathbb{E}[(\mathbf{\Phi} \mathbf{w})(\mathbf{\Phi} \mathbf{w})^\mathsf{T}] = \mathbf{\Phi} \mathbb{E}[\mathbf{w} \mathbf{w}^\mathsf{T}] \mathbf{\Phi}^\mathsf{T} = \lambda^2 \mathbf{\Phi} \mathbf{\Phi}^\mathsf{T}$. As a result, the distribution of $\mathbf{y}$ becomes a multivariate Gaussian distribution, $\mathbf{y} \sim \mathcal{N}(\mathbf{0}, \lambda^2 \mathbf{\Phi} \mathbf{\Phi}^\mathsf{T})$. By defining the covariance matrix as $\mathbf{K} = \lambda^2 \mathbf{\Phi} \mathbf{\Phi}^\mathsf{T}$, the $(n, n')$

elements become $k(\mathbf{x}_n, \mathbf{x}_{n'}) = \phi(\mathbf{x}_n)^\mathsf{T}\phi(\mathbf{x}_{n'})$. Now, by constructing the kernel matrix $\mathbf{K}$ by directly defining the kernel function $k(\mathbf{x}_n, \mathbf{x}_{n'})$, there's no need to explicitly define the feature vector $\phi(\mathbf{x})$ (kernel trick). Here, the definition of the GPs is that for any set of $N$ inputs $(\mathbf{x}_1, \ldots, \mathbf{x}_N)$, if the joint distribution $p(\mathbf{y})$ of the corresponding outputs $\mathbf{y} = (y_1, \ldots, y_N)$ follows a multivariate Gaussian distribution, the relationship between $\mathbf{x}$ and $y$ is governed by a GPs. Now, $\mathbf{y} \sim \mathcal{N}(\mathbf{0}, \mathbf{K})$ is a GPs with mean $\mathbf{0}$ and covariance matrix $\mathbf{K}$. It should be noted that we can centered the mean of the observed data $\mathbf{y}$ at $\mathbf{0}$ without loosing generalization. For the training data $\mathbf{X}^N = (\mathbf{x}_1, \ldots, \mathbf{x}_N)$ and $\mathbf{y}^\mathbf{N} = (y_1, \ldots, y_N)^\mathsf{T}$, and the data we wish to predict $\bar{\mathbf{X}}^S = (\bar{\mathbf{x}}_1, \ldots, \bar{\mathbf{x}}_S)$, the joint distribution of the corresponding output $\bar{\mathbf{y}}^S = (\bar{y}_1, \ldots, \bar{y}_S)^\mathsf{T}$ is given by:

$$
\begin{pmatrix} y_1 \\ \vdots \\ y_N \\ \bar{y}_1 \\ \vdots \\ \bar{y}_S \end{pmatrix} \sim \mathcal{N} \left( \begin{pmatrix} 0 \\ \vdots \\ 0 \\ 0 \\ \vdots \\ 0 \end{pmatrix}, \begin{array}{c} \mathbf{x}_1 \\ \vdots \\ \mathbf{x}_N \\ \bar{\mathbf{x}}_1 \\ \vdots \\ \bar{\mathbf{x}}_S \end{array} \left( \begin{array}{c|c} \mathbf{K}_N + \sigma^2\mathbf{I} & \bar{\mathbf{K}}_{S,N} \\ \hline \bar{\mathbf{K}}_{S,N}^\mathsf{T} & \bar{\mathbf{K}}_S \end{array} \right) \right).
\tag{23}
$$

$\sigma^2\mathbf{I}$ represents the variance of observational noise, modeling the presence of noise in the training data. The predictive distribution can be analytically derived as

$$
p(\bar{\mathbf{y}}^S | \bar{\mathbf{X}}^S, \mathbf{y}^N, \mathbf{X}^N) = \mathcal{N}(\bar{\mathbf{K}}_{S,N}^\mathsf{T}[\mathbf{K}_N + \sigma^2\mathbf{I}]^{-1}\mathbf{y}^N, \bar{\mathbf{K}}_S - \bar{\mathbf{K}}_{S,N}^\mathsf{T}[\mathbf{K}_N + \sigma^2\mathbf{I}]^{-1}\bar{\mathbf{K}}_{S,N}).
\tag{24}
$$

