# OpenReview forum: "Fast Computation of Gaussian Processes Augmented by Synthetic Simulator Data"
_ICLR.cc/2025/Conference — ICLR 2025 Conference Withdrawn Submission_

### Official Review · Reviewer_XBpK · 2024-10-17

**Soundness:** 1
**Presentation:** 1
**Contribution:** 1
**Rating:** 3
**Confidence:** 4

**Summary:**

This paper considers the problem of augmenting Gaussian process regression methods with synthetic data generated from a simulator. This simulator may be misspecified, in that the distribution of synthetic data differs from the distribution of data generated from the true underlying data generating process. As a result, it may not be desirable to include all of the synthetic data in the Gaussian process fit, and the authors propose an approach to select which synthetic data points to include using the marginal likelihood.

**Strengths:**

There are not many strengths to this paper. One strength is that the authors thought carefully about the computational cost of their method and how to reduce this where possible.

**Weaknesses:**

- Writing: The paper is very poorly written: (i) the introduction feels like disjoint sentences pasted together rather than coherent paragraphs, (ii) the paper lacks a proper background section on Gaussian processes to set notation and terminology, which makes the rest of the paper confusing (despite the fact I have written 20+ papers on Gaussian processes), (iii) the methodology sections (i.e. section 3 and 4) need to be restructured to present a clear set of equations which are implemented, rather than vague and long derivations which are unnecessary, (iv) the experiments section provides very little detail on competitors and the figures could do with clear captions and much more discussion of the results.

- Mathematics: The paper both lacks mathematical details and provides too many. It lacks them in the background, where setting notation in a clear and concise way could be extremely helpful. However, there is way too much detail in Section 4. I would recommend that the authors rewrite section 4 in a much more concise manner and make use of lemmas/propositions/theorems to hide unnecessary derivations. Even in cases where mathematics is defined, such as on lines 126-133, there seems to be confusions/inaccuracies (in this case with respect to the difference between random variables and draws from random variables). In summary, the mathematical details in this paper need some very serious polishing.

- Baselines: The authors seem to be unaware of a large part of the literature in this field. For example, an obvious approach to take here would be transfer learning (specifically multi-task learning) where the real and simulate data sets are seen as functions which can be jointly modelled to encode the relationship/correlation between them. Alternatively, multi-fidelity modelling would also be appropriate; the synthetic data could be considered as low-fidelity data to complement the real data (which would be considered as high-fidelity). The absence of discussion and thorough comparison with these baselines makes the paper severely flawed and not suitable for publication. I would recommend that the authors look at the following papers and related literature:

Bonilla, E., Chai, K. M., & Williams, C. (2008). Multi-task Gaussian Process Prediction. Advances in Neural Information Processing Systems, 20(October), 153–160.

Perdikaris, P., Raissi, M., Damianou, A., Lawrence, N. D., & Karniadakis, G. E. (2016). Nonlinear information fusion algorithms for robust multi-fidelity modeling. Proceedings of the Royal Society A: Mathematical, Physical, and Engineering Sciences, 473(2198).

Parussini, L., Venturi, D., Perdikaris, P., & Karniadakis, G. E. (2017). Multi-fidelity Gaussian process regression for prediction of random fields. Journal of Computational Physics, 336, 36–50. https://doi.org/10.1016/j.jcp.2017.01.047

Peherstorfer, B., Willcox, K., & Gunzburger, M. (2018). Survey of multifidelity methods in uncertainty propagation, inference, and optimization. SIAM Review, 60(3), 550–591.

Minor points:

- What is a "Plane" GP? Do you mean "plain"? If so, what is a "plain" GP?
- Please look into using \citep instead of \cite where appropriate.

**Questions:**

See the weaknesses section above. Realistically, there is very little the authors can do to convince me to change my mind on this paper without seeing an entirely new draft.

---

### Official Review · Reviewer_TY8t · 2024-10-18

**Soundness:** 1
**Presentation:** 2
**Contribution:** 1
**Rating:** 3
**Confidence:** 5

**Summary:**

This article considers the setup when existing data can be complemented by additional simulated data. The question is then to select new simulated data to improve a Gaussian process surrogate. The criterion to do so is the log-marginal likelihood, helped by matrix update formulas.

**Strengths:**

The paper connects with anomaly detection and sparse GP regression.

**Weaknesses:**

Unfortunately, it seems that the authors are not aware of the literature on multi-fidelity modeling with Gaussian processes, which deals exactly with the question of combining data of different accuracy. Here the model mixes both fidelity levels, which is not appropriate. Some initial references to multi-fidelity, with many references therein:
- Kennedy, M. C., & O'Hagan, A. (2000). Predicting the output from a complex computer code when fast approximations are available. Biometrika, 87(1), 1-13.
- Garnett, R. (2023). Bayesian optimization. Cambridge University Press.
- Le Gratiet, L., & Cannamela, C. (2015). Cokriging-based sequential design strategies using fast cross-validation techniques for multi-fidelity computer codes. Technometrics, 57(3), 418-427.

Then it seems that the authors details the update of the predictive variance, which is well known from update formulas. See / check / refer to, e.g.,:
- Chevalier, C., Ginsbourger, D., & Emery, X. (2013, October). Corrected kriging update formulae for batch-sequential data assimilation. In Mathematics of Planet Earth: Proceedings of the 15th Annual Conference of the International Association for Mathematical Geosciences (pp. 119-122). Berlin, Heidelberg: Springer Berlin Heidelberg.
- Gramacy, R. B. (2020). Surrogates: Gaussian process modeling, design, and optimization for the applied sciences. Chapman and Hall/CRC.

**Questions:**

Other related reference: Mak, S., & Joseph, V. R. (2017). Projected support points: a new method for high-dimensional data reduction. arXiv preprint arXiv:1708.06897.

Plane GP: do you mean plain/vanilla GP?

---

### Official Review · Reviewer_ECkq · 2024-10-29

**Soundness:** 3
**Presentation:** 3
**Contribution:** 3
**Rating:** 6
**Confidence:** 2

**Summary:**

This article proposes a new method for using simulated data alongside observed data to train Gaussian Processes models. The article proposes using the log marginal likelihood as a criterion for accepting the simulated data, in the presence of a potentially misspecified simulator where the samples from the simulator cannot all be assumed to be representative of the real data generating function. The use of this acceptance criterion is then demonstrated to improve the predictive performance, when compared to just using the observational data or other data-augmentation methods.

**Strengths:**

The paper is generally fairly easy to understand and conceptually well-motivated. The experiments are pretty thorough methodologically: more than 10 iterations would be nice, although I suppose training a bunch of GPs over and over again will be expensive, and I appreciate the characterisation of the computational costs. The treatment of misspecification in the experiments is clear to understand while also being realistic.

**Weaknesses:**

There is one major and minor issue with the mathematical notation used:

Section 4 contains some clear notational mistakes: you write K_{m+1*}^{-1} + sigma^2I when you mean (K_{m+1*} + sigma^2I )^{-1} at least four times. This is a serious error, but I’m pretty sure it’s just a mistake in the notation and not the methods.

The footnote notation “(x,y)2” is a bit confusing.

There are a few small English grammar issues:

“Contributions of this research are 2 folds. First, we proposed…”

“SoD” acronym is used after not being mentioned for several pages.

“Plane GP” Do you mean “Plain GP”? As in, by itself? Text and tables figures need changing.

“The lax assumptions of the GPs”: “lax” is not quite the right word here. “Weak”, maybe?

“The correct knowledge” from the simulator: well, in principle all knowledge is correct. “Information” might be a better word.
Little typo in appendix C: “lengthscale”

There are a couple of small issues with the diagrams:

Mysterious white space top of page 10

Ablation study a good idea in Appendix E. Figure 5 could be better quality, more in the style of Figure 1.

The literature review descriptions of sparse GPs are not strictly accurate:

“Two approaches have been developed to reduce the amount of training data… Sparse Gaussian processes, which generate a small number of pseudo-training data points”: this is not strictly right: sparse GPs use as much training data as a full GP model, but they project the information in the training data through the inducing points, leading to computational speedups.

Appendix A.2. Again your characterisation of Sparse GPs is kind of strange. All of the training data is available for them, but log marginal likelihood is not computable because it is too computationally expensive,so they pursue a variational approach to lower bound it and optimise against that instead. The marginal likelihood isn’t altered, it’s just not accessible.

**Questions:**

Model specification questions:

“The data generated by simulators often deviates from the true distributions”: Do you not always assume this to be true, rather than just “often”? “All models are wrong…” and all that.
There is always the issue of the GP itself being misspecified: one expects a nonparametric approach to minimise these risks but there are always concerning with kernel choice, and GPs being overly smooth, and having misspecified noise processes, to name a few. Does the principle of of using the log-marginal likelihood to assess the simulator quality hold if the GP is misspecified? What if the simulator is better-specified than the GP?

Numerical questions:

Concerning the computational trick of the rank-one updates: what are the concerns around acquiring numerical error by iteratively reusing the values from the previous step? Consider Algorithm 3.5 in Rasmussen and Williams book where this issue is mentioned and corrected for.

Observed/simulated data sharing questions:

Are simulated data treated exactly the same as observed data when training the GP? If so, is this wise?
Might you end up in a situation where the simulated data dominates the observed data and the inclusion criteria deteriorate?
Could these be integrated into one model while still modelling and acknowledging the differences? Something like multi-output regression could work here to learn related but non-identical target variables.
How is the simulator trained? Using other data independent of the data used for the GP? Using expert elicitation?
If you had a very small amount of real observed data, would you lower the threshold for accepting simulated data since you needed more of it? Or raise the threshold to ensure that the accepted simulated data doesn’t dominate too much?

Computational questions

To what extent does the computational cost of generated simulated data influence your method? Would an expensive simulator cause you to lower the threshold for accepting points?
It is good to see computational time presented in section 5.3 or table 4, but does the measured time match the big-O predictions? Can you plot this? Would a version of figure 4 with a log-transformed y axis be clearer?

Experiment methodological questions:

“Termination condition was set to accept until half of all simulator data were accepted” Why half?
Would metrics other than the MSE be interesting? Are the MSE and the logML so closely related for the GP that it doesn’t really count as an independent metric? Did you consider using anything else?
Appendix C: How did you choose the parameters for training the data if you didn’t optimise them?
Why don’t you relearn the hyperparameters after algorithm 1? This seems like it would be a good idea. Bayesian optimisation algorithms will relearn the hyperparameters every few acquisition steps just to reflect the influence of the newly included data. Could you sample the kernel hyperparameters rather than optimise them? Would this give a better characterisation of the logML target and choose better simulated data?

---

### Official Review · Reviewer_Vrax · 2024-11-01

**Soundness:** 2
**Presentation:** 1
**Contribution:** 1
**Rating:** 3
**Confidence:** 4

**Summary:**

Gaussian processes are prized for being data-efficient models that do not require large datasets for training. Nevertheless, data augmentation can still improve overall performance if additional data-points can be acquired from either the true data source or a simulation. In this paper, the authors present an approach for selecting datapoints using the log marginal likelihood as the selection criteria for either adding a new datapoint to the training set or discarding it. The drawback of this approach is that computing the log marginal likelihood has $O(N^3)$ time complexity, so the authors propose a solution using rank-one updates to the Cholesky factorization. The effectiveness of the approach is evaluated for GP regression using a mixture of synthetic and public datasets to emulate the set-up of having simulated and true distributions of data.

**Strengths:**

- Investigating solutions for combining limited data from a true distribution with more accessible data points available via simulation is an interesting problem having various use-cases in industry. However, I am not convinced that the problem statement is presented in a compelling enough manner in the paper (see Weaknesses).

- Leveraging the log marginal likelihood as the selection criteria seems to be an appropriate choice, and the efforts to address the infeasible time complexity associated with standard Cholesky decomposition poses an interesting research problem that is widely explored in the literature.

**Weaknesses:**

- The motivation for the paper comes across as weak, and I couldn’t understand why there were no connections to multi-fidelity modelling or active learning. The former is directly applicable to scenarios where data come from multiple sources of fidelity, as is the case here with simulator-generated data versus real-world training data. Multi-fidelity GPs explicitly handle such cases by blending high- and low-fidelity data, allowing the model to balance simulator data with real-world data. I consider active learning to also be relevant here - the authors’ proposed approach involves deciding which simulator-generated points to add based on the log marginal likelihood, but active learning methods are also concerned with this data selection strategy.

- Beyond the connections suggested above, there appears to be a large disconnect of the paper from existing literature. Besides the citation for the `pvlib` python library, all other cited papers were published before 2021. While earlier work is undoubtedly foundational, this still presents a clear gap in the reviewed literature. Even in this light, the paper’s contributions appear to be fairly incremental.

- I unfortunately found the paper to be quite poorly-written overall. It contains several typos and inconsistencies that could have been identified with more thorough proof-reading before submission. I don’t think the presentation quality meets the standard expected for ICLR in its current form.

**Questions:**

See Weaknesses section above.

---

### Note · Authors · 2024-11-20

**Comment:**

As most reviewers pointed out, a comparison with Multi-fidelity GPs is necessary, and since it takes time, we will withdraw.

**Withdrawal Confirmation:**

I have read and agree with the venue's withdrawal policy on behalf of myself and my co-authors.